# Comparing the Risk of Infusion-Related Reactions and Tolerability in Patients Given Cetirizine or Diphenhydramine Prior to Ocrelizumab Infusion (PRECEPT)

**DOI:** 10.3390/medicina60040659

**Published:** 2024-04-19

**Authors:** Kyle Smoot, Horia Marginean, Tiffany Gervasi-Follmar, Chiayi Chen

**Affiliations:** Providence Brain and Spine Institute, Portland, OR 97225, USA; horia.marginean@providence.org (H.M.); tiffany.gervasi-follmar@providence.org (T.G.-F.); chiayi.chen@providence.org (C.C.)

**Keywords:** multiple sclerosis, ocrelizumab, infusion reactions, pre-medication, patient reported outcomes

## Abstract

*Background:* Ocrelizumab is an effective medication for multiple sclerosis. However, infusion-related reactions (IRRs) are a concern for patients and may lead to discontinuation of ocrelizumab. To minimize IRRs, pre-medications are administered. However, from our experience, these medications, especially diphenhydramine, can cause marked drowsiness. The primary objective of this study was to evaluate whether cetirizine is non-inferior to diphenhydramine in limiting the proportion and severity of reactions from ocrelizumab infusions. *Methods:* Twenty participants were serially randomized in a 1:1 ratio to receive 10 mg of cetirizine or 25 mg of diphenhydramine orally prior to their first three ocrelizumab infusions. *Results:* The rate of IRRs in this study was similar across both treatment groups with no increase in the risk of severity, and no grade 3 IRRs. Further, patients receiving cetirizine experienced a reduction in fatigue. While there was not a significant difference in global satisfaction, this score increased over time in the cetirizine arm while it remained unchanged in the diphenhydramine arm. *Conclusions:* Overall, our results suggest that cetirizine does not increase the risk of infusion-related reactions compared to diphenhydramine.

## 1. Introduction

Ocrelizumab (OCR), a humanized monoclonal antibody that selectively binds to and destroys CD20-expressing lymphocytes, has demonstrated robust efficacy in reducing relapse rate, sustained disability worsening, and MRI indicators of disease activity in patients with relapsing multiple sclerosis (RMS) [1]. For patients with primary progressive MS (PPMS), ocrelizumab demonstrated a reduction in disability progression compared to patients receiving the placebo treatment [2]. Moreover, ocrelizumab is well tolerated. Upper respiratory infections are the most common potential side effect of ocrelizumab, followed by infusion-related reactions (IRRs). More than 34 percent of patients with RMS in controlled ocrelizumab trials experienced at least one IRR (OPERA I and OPERA II trials) [1]. Most IRRs were mild to moderate and occurred with the first infusion of dose one. In these clinical trials, all patients were pre-medicated with intravenous methylprednisolone, an analgesic/antipyretic, and an intravenous or oral antihistamine, such as diphenhydramine, to minimize IRRs. However, in our clinic, we observed that patients experienced marked sedation, and subsequent impairment in their ability to function, drive, or work, in some cases for 24 h after completion of the infusion. As a result, we decreased the dose of diphenhydramine to 25 mg, administered either by IV or orally. However, patients continued to experience drowsiness. It is for this reason that we sought to explore the use of another antihistamine that would be as effective in reducing infusion reactions but without producing the unwanted side effects associated with the use of diphenhydramine. Cetirizine has been reported to be less sedating than diphenhydramine [3]. In several oncology studies, cetirizine was demonstrated to be less sedating with infrequent infusion reactions [4]. Intravenous (IV) administration of 10 mg of cetirizine compared to IV administration of 50 mg of diphenhydramine was as effective at preventing IRRs and more tolerable in a small study of 25 patients who received rituximab for the treatment of hematologic or solid tumor malignancies [5]. In addition, pre-medication with 10 mg of cetirizine the night prior to the infusion has been reported to reduce IRRs [6]. However, we were unaware of any evidence regarding the efficacy of oral cetirizine compared to oral diphenhydramine in reducing IRRs to ocrelizumab in persons with multiple sclerosis. The primary objective of this study was to evaluate whether cetirizine is non-inferior to diphenhydramine in limiting the proportion and severity of reactions from ocrelizumab infusions. The secondary objective was to evaluate patient treatment satisfaction after receiving cetirizine and diphenhydramine as pre-medication for ocrelizumab infusions.

## 2. Methods

### 2.1. Trial Design

This was a single-center, randomized, parallel arm, comparative study of the risk and severity of infusion reactions to ocrelizumab following pretreatment with cetirizine versus diphenhydramine. Twenty participants were serially randomized in a 1:1 ratio to receive 10 mg of cetirizine orally or 25 mg of diphenhydramine orally prior to ocrelizumab infusions on day 0 (first infusion of the first dose), day 14 (second infusion of the first dose), and day 168 (second dose) to determine whether there was a difference in tolerance without an increase in ocrelizumab IRRs. Study participation required four visits (screening, baseline, day 14, and day 168) and lasted about 6 months (Figure 1).

### 2.2. Participants and Setting

Patients were considered for the study if they were eligible to receive ocrelizumab according to FDA-approved indications for MS treatment [7]. Inclusion criteria included a diagnosis of a relapsing or progressive form of MS; age of 18 to 70 years inclusive at the time of consent; naïve to ocrelizumab treatment; and an expanded disability status scale (EDSS) of ≤6.5 at screening. Female patients of childbearing potential were required to agree to practice effective contraception and continue contraception during the study. All patients were seen by an MS specialist at the Providence MS clinic for the duration of the study.

Exclusion criteria were a diagnosis of mental conditions that potentially prevented the patient from understanding the nature, scope, and possible consequences of the study, or that risked the patient not adhering to the study protocol; evidence of active hepatitis B infection at screening; untreated hepatitis C or tuberculosis infection; a history of progressive multifocal leukoencephalopathy (PML); a history of ever testing positive for HIV; persistent or severe infection; pregnancy or lactation; significant, uncontrolled somatic disease or severe depression in the last year; concurrent use of immunosuppressive medication, lymphocyte-depleting agents, or lymphocyte-trafficking blockers; significant comorbidity that, in the opinion of the investigator, would interfere with participation in the study; and allergy or inability to tolerate diphenhydramine or cetirizine.

Participants were recruited between 5 February 2020 and 30 November 2021 from the Providence Brain and Spine Institute, Portland, OR, USA. The last day on which treatment was administered for the study was 13 May 2022.

### 2.3. Interventions

An amount of 10 mg of cetirizine or 25 mg of diphenhydramine was administered orally 30–60 min prior to each ocrelizumab infusion. Dose modifications of diphenhydramine or cetirizine were not allowed. In addition, each participant received the same dose of acetaminophen and methylprednisolone.

Ocrelizumab was administered according to the standard of care in an approved hospital or outpatient infusion center under close supervision of the investigator or medically qualified staff. The treatment regimen was 300 mg IV on day 0, 300 mg IV on day 14 ± 2 days, and 600 mg IV on day 168 ± 14 days (Figure 1).

### 2.4. Variables and Outcomes

The primary outcome of the study was the proportion of patients with an infusion-related reaction during or after the first infusion of the first ocrelizumab dose of each pre-medication arm.

Secondary outcomes included the following: the proportion of patients with an infusion-related reaction during or after receiving the second infusion of the first dose on day 14 and the infusion of the second dose on day 168; the proportion of patients that required ocrelizumab dose adjustments as a consequence of infusion-related reactions; and changes in treatment satisfaction questionnaire for medication (TSQM), Stanford sleepiness scale (SSS), visual analog scale for fatigue (VAS-F), modified fatigue impact scale (MFIS), and multiple sclerosis impact scale (MSIS-29, physical and psychological) scores across the four study visits (Figure 1).

Demographic and clinical characteristics were abstracted from electronic medical records and confirmed at the screening visit.

#### 2.4.1. Adverse Events

Adverse events (AEs) were defined as unfavorable unintended signs, symptoms, or diseases that were experienced by the patient, whether or not there was thought to be a causal relationship with either cetirizine or diphenhydramine. Adverse events (AEs) that occurred after consent and up to 30 days following the last ocrelizumab administration during this study, or study discontinuation or termination, whichever was earlier, were reported. AEs were graded according to the National Cancer Institute Common Terminology Criteria for Adverse Events (CTCAE) version 5.0. The timing, severity, potential relationship to ocrelizumab and the pre-medication, and degree of seriousness of AEs were recorded. AEs reported during or within 24 h after ocrelizumab infusion were assessed by the investigator to determine if they were infusion-related reactions (IRRs). All study AEs and serious AEs (SAE) were reported according to the International Conference of Harmonization and local IRB guidelines.

#### 2.4.2. Neurological Exams and Patient-Reported Outcomes

Full neurological exams and the estimated expanded disability status scale (EDSS) assessment were conducted at screening and day 168, or at premature termination, if applicable.

The Stanford sleepiness scale (SSS) and the visual analog scale for fatigue (VAS-F) questionnaires were administered at screening, day 0, day 14, and day 168. A higher VAS-F score indicates greater fatigue. A higher SSS score is associated with increased sleepiness.

The modified fatigue impact scale (MFIS) and multiple sclerosis impact scale (MSIS-29) questionnaires were administered at screening, at day 168, or at premature termination, if applicable. 

A treatment satisfaction questionnaire for medication (TSQM) was administered on day 0, day 14, and day 168. TSQM, SSS, and VAS-F questionnaires were administered within 2 h after each ocrelizumab infusion, followed by a phone call on the next business day following each infusion to collect and assess infusion reactions or other AEs.

### 2.5. Sample Size

Based on the results of the OPERA I and II trials, we assumed that 25% of patients in the diphenhydramine arm and 25% of patients in the cetirizine arm would experience mild to severe IRRs related to the first infusion on day 0. Further, we assumed that a 30% absolute difference in the proportion of patients with IRRs between pre-medication groups is the maximum clinically relevant difference, assuming that cetirizine is not inferior to diphenhydramine. To achieve 80% power with a one-sided significance level of 0.05, the study required 26 patients per arm, for a total of 52 participants. However, the study did not meet the enrollment. The main reason for reduced enrollment was the SARS-CoV-2 pandemic since the trial required additional time and longer visits. In addition, several patients were infused off site due to their insurance, so we were not able to capture the patient-reported outcomes.

Diphenhydramine or cetirizine was discontinued if the patient experienced a life-threatening or serious hypersensitivity reaction, discontinued ocrelizumab therapy, or elected to discontinue therapy for any reason. Ocrelizumab was discontinued if the patient experienced a life-threatening IRR or serious hypersensitivity reaction, tested positive for an active hepatitis B infection, PML, tuberculosis, HIV, became pregnant, had unacceptable toxicity, multiple serious infections, or elected to discontinue therapy for any reason.

### 2.6. Randomization, Allocation, and Blinding

Randomization was based on a computer-generated sequence using a permuted block design with a block size equal to four. Patients were assigned in a 1:1 ratio to oral diphenhydramine and oral cetirizine. There was no blinding of the pre-medication. The treating physician prescribed the pre-medication according to the prescribing instructions provided by the drug manufacturer.

### 2.7. Statistical Methods

The demographics and clinical characteristics are summarized as medians (interquartile range [IQR]) for continuous variables and as frequencies and percentages for categorical variables.

Since the enrollment goals were not reached, the randomization sequence was not completed as per protocol. As a quality control measure, we compared baseline demographic and clinical characteristics between the two treatment groups to assess the success of randomization.

Continuous variables were compared using the Wilcoxon rank sum test. Categorical variables were compared with Fisher’s exact test.

Analyses were based on the intention-to-treat (ITT) principle and were performed using all patients who were randomized and received at least one dose of either oral diphenhydramine or oral cetirizine.

The primary efficacy outcome, the proportion of patients with IRRs after the first infusion of dose 1, was reported for each arm during or after the first infusion on day 0 and compared using Fisher’s exact test. The secondary outcomes, the proportion of patients with IRRs after the second infusion of dose 1 and dose 2, were reported for each arm at day 14 and day 168 and compared using Fisher’s exact test.

TSQM scores were compared between arms at day 0, day 14, and day 168 using the Wilcoxon rank sum test. SSS and VAS-F scores were compared using the Wilcoxon rank sum test. MFIS and MSIS-29 (physical and psychological) scores were compared on day 168 using multiple linear regression with analysis of covariance (ANCOVA), adjusting for the screening score. For TSQM, SSS, VAS-F, MFIS, and MSIS-29 analyses, patients with missed assessments due to early termination or unavailability were classified as non-responders.

All reported *p*-values were 2-tailed, and *p*-values less than 0.05 were considered statistically significant. Analyses comparing pre-medication groups were conducted using R statistical software (R version 4.0.4 (15 February 2021), the R Foundation for Statistical Computing Platform).

## 3. Results

The required sample size of 52 patients was not achieved. Twenty MS patients were screened, but one patient did not meet the inclusion criteria because of previous treatment with ocrelizumab. A total of 19 participants were randomized (cetirizine, n = 10; diphenhydramine, n = 9) and completed a day 0 baseline visit (Figure 2). Study enrollment took 22 months between 5 February 2020 and 30 November 2021. The recruitment rate was fewer than one participant per month. The major obstacle to enrollment into the study was the SARS-CoV-2 pandemic.

Among those enrolled, 18/19 completed the study. One participant withdrew consent after the second infusion because of an insurance change. Baseline demographics and clinical characteristics were similar between treatment groups (Table 1). Median patient follow-up for the study was 7.1 months (95% CI, 6.67, 7.99).

### 3.1. Treatment Outcomes

We did not achieve the pre-scheduled sample size for the non-inferiority analysis in the intention to treat population (ITT). We cannot prove that cetirizine was not inferior to diphenhydramine in reducing IRRs after the first 300 mg ocrelizumab infusion. 

After the first infusion of dose one, six patients in each study arm experienced IRRs (60% of patients treated with cetirizine and 67% of patients treated with diphenhydramine, Figure 3). After the second infusion of dose one, the number of IRRs decreased to 5/10 (50%) patients treated with cetirizine and 4/9 (44%) patients treated with diphenhydramine. At the end of the study, 8/10 patients (80%) treated with cetirizine and 8/9 (89%) patients treated with diphenhydramine experienced at least one IRR (Figure 3). The rate of IRRs between the two treatment groups was not significantly different.

Mean (standard deviation [SD]) baseline VAS-F fatigue and energy domains were similar in both treatment groups: (CTZ, 2.36 [2.50]; DPH, 3.13 [1.65]) and (CTZ, 6.02 [2.80]; DPH, 5.36 [2.34]). There was a significant difference in the VAS-F fatigue domain after the first two 300 mg OCR infusions favoring patients pre-treated with cetirizine (*p* = 0.03 and *p* = 0.04), but no difference in the VAS-F energy domain (Table 2).

Patients on cetirizine showed significant improvement in SSS between the screening visit and the last visit after the second ocrelizumab dose on day 168 (Table 2, *p* = 0.03). MFIS and MSIS-29 scores at screening were higher in the patients randomized to diphenhydramine, with no change after the second dose (Table 3). However, MFIS and MSIS-29 scores improved in both groups after their second dose of ocrelizumab compared to screening.

After adjusting for the baseline scores, there was no significant difference between the cetirizine and diphenhydramine groups following the second ocrelizumab infusion in the TSQM global satisfaction scores or the subscales of effectiveness, side effects, and convenience (Table 4).

### 3.2. Incidence of Adverse Events

The incidence of adverse events was balanced between groups. There were no grade 3 or higher adverse events in either group. One serious adverse event (COVID-19 infection) was reported in the diphenhydramine group, but it was not deemed related to diphenhydramine. No patients discontinued due to adverse events.

## 4. Discussion

Ocrelizumab is an effective medication for MS, but IRRs were reported between 34.3% in the pooler OPERA trials and 39.9% in the ORATORIO trial [8]. While most IRRs are mild, they are still a concern for patients and may lead to discontinuation of ocrelizumab. IRRs can consist of multiple symptoms including fever, chills, and myalgias, and they typically occur within the first hour of the infusion. These symptoms are more prominent early in the treatment course due to the presence of B cells, and when B cells are rapidly depleted this results in a cytokine release [9]. To minimize IRRs, pre-medications are administered. However, from our experience, these medications, especially diphenhydramine, can cause marked drowsiness. Reducing post-infusion drowsiness is important because it can increase the risk of falls or other trauma, and patients may not be safe to drive after an infusion. In addition, patients who experience sedation may need to spend more time under observation at an infusion center. Cetirizine is known to cause less drowsiness compared to diphenhydramine, and several studies in patients receiving treatment for hematologic or solid tumor malignancies pre-medicated with cetirizine have demonstrated fewer IRRs while improving tolerability [3]. Therefore, we undertook this study to compare the efficacy and side effect profile of diphenhydramine and cetirizine.

The rate of IRRs in this study was similar across both treatment groups with no increase in the risk of severity, and no grade 3 IRRs [10]. As seen in the phase III clinical trials, the rate of infusion reactions in our study decreased over time [1,2]. While the percentage of patients who experienced infusion reactions was elevated in this trial compared to the phase III trials, a direct comparison is challenging [11]. Phase III clinical trials typically enroll a younger group of patients with less co-morbidities. For example, in the OPERA I and II trials, patients older than 55 were excluded from enrolling. In our study, the mean age was 48 years compared to 37 years in the phase III trials. Also, three patients were older than 55. In a previous review of rituximab which has a similar mechanism of action as ocrelizumab, older patients were more likely to experience IRRs [12]. In addition, the investigators and patients were not blinded to the treatment used to possibly prevent IRRs. Also, since IRRs were the primary outcome of this study, there may have been more vigilance in capturing these reactions.

Patients receiving cetirizine experienced a reduction in fatigue as measured by the VAS-F and SSS questionnaires. While there was not a significant difference in the global satisfaction of the TSQM results, this score increased over time in the cetirizine arm while it remained unchanged in the diphenhydramine arm. We observed a similar tendency in the TSQM subset scores.

Not surprisingly, there was no significant difference in the MFIS and MSIS-29 scores between the screening visit and the second dose of ocrelizumab. These scales are probably not sensitive enough to capture a difference in pre-medications for the infusions. However, the scores did decrease in both treatment groups, possibly indicating a beneficial effect due to ocrelizumab.

## 5. Conclusions

Overall, our results suggest that cetirizine does not increase the risk of infusion-related reactions compared to diphenhydramine. However, one limitation of this study was the small number of patients enrolled in the trial, which was mostly due to the COVID-19 pandemic. In addition, many of our patients were not able to receive their ocrelizumab infusions at our facility due to insurance restrictions. Therefore, we could not enroll them. Potentially, if our sample size was larger, we would have seen an impact on the quality-of-life measures. Another limitation is that patients and researchers were not blinded to the treatment arm, which may have created bias. Nonetheless, this is the only study we are aware of that demonstrates tolerability and safety using cetirizine instead of diphenhydramine as a pre-medication prior to patients with MS receiving ocrelizumab. Performing a study with a larger patient population would possibly result in more significant results.

## Figures and Tables

**Figure 1 medicina-60-00659-f001:**
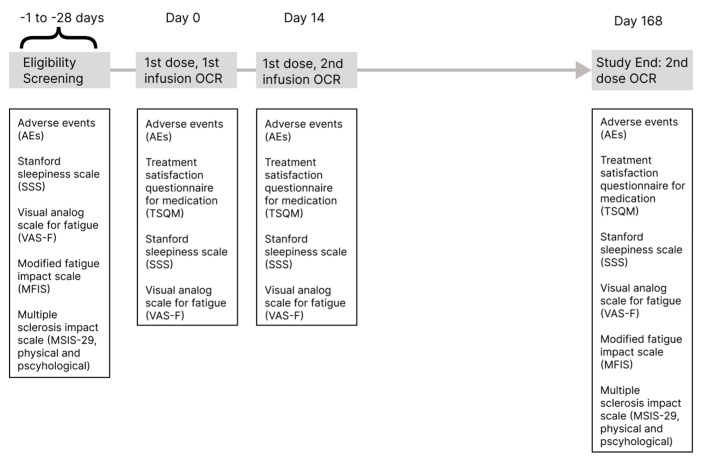
Study timeline. This diagram displays the timeline of study visits relative to the first infusion of the first dose of ocrelizumab, which was administered on day 0. Eligibility screening took place between day-1 and day-28. The second infusion of the first dose of ocrelizumab occurred on day 14. The second and final dose of ocrelizumab occurred on day 168.

**Figure 2 medicina-60-00659-f002:**
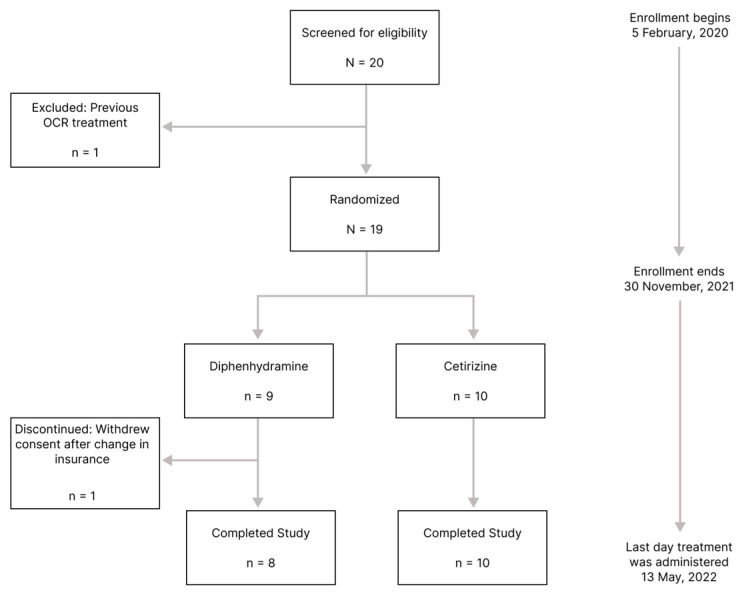
Consort diagram. This diagram illustrates the participant recruitment and screening process, including reasons potential participants were excluded from the study and reasons for discontinuing the study.

**Figure 3 medicina-60-00659-f003:**
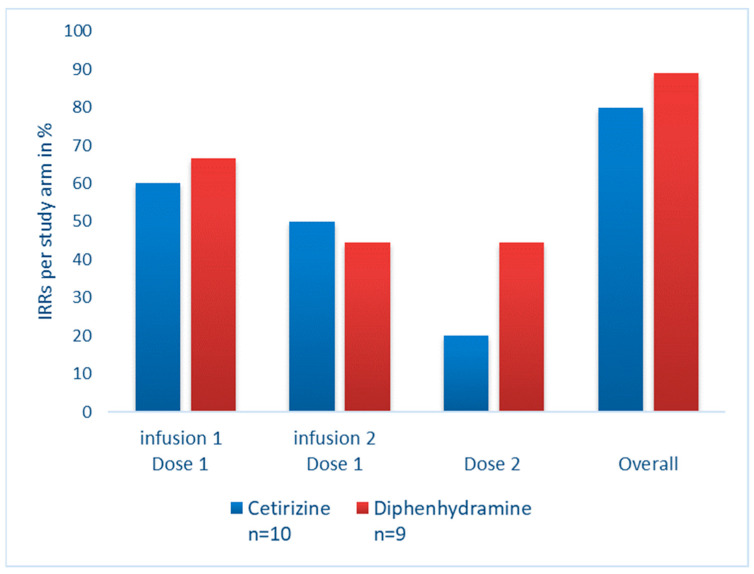
Infusion-related reactions after each infusion by antihistamine treatment arm. This bar graph shows the percentage of patients who experienced infusion-related reactions after each infusion of ocrelizumab. Patients in the cetirizine arm are represented by a blue bar. Patients in the diphenhydramine arm are represented by a red bar.

**Table 1 medicina-60-00659-t001:** Demographics of progressive MS patients by arm.

			Arm		
Relapsing or Progressive MS		Total (N = 19) ^1^	Cetirizine (n = 10, 53%) ^1^	Diphenhydramine (n = 9, 47%) ^1^	*p*-Value ^2^
Age at MS symptoms onset (years)		33.9 [19.4, 56.4]	33.7 [29.5, 47.0]	33.9 [30.0, 36.5]	0.87
Age at MS diagnosis (years)		34.6 [19.8, 63.1]	34.5 [29.6, 47.1]	36.7 [30.1, 45.3]	0.99
From MS symptoms onset to MS diagnosis (weeks)		8.6 [0, 465]	7.4 [2.5, 25.9]	8.6 [5.4, 209]	0.27
Age at OCR start (years)		47.5 [29.0, 63.2]	48.2 [38.0, 54.0]	46.3 [39.8, 52.2]	0.97
Gender, n (%)	Female	15 (78.9%)	9 (90.0%)	6 (66.7%)	0.30
Race, n (%)	Asian	1 (5.3%)	1 (10.0%)	0 (0.0%)	0.99
	Black or African American	2 (10.5%)	1 (10.0%)	1 (11.1%)	
	Other	1 (5.3%)	0 (0%)	1 (11.1%)	
	White	15 (78.9%)	8 (80.0%)	7 (77.8%)	
Ethnicity, n (%)	Hispanic or Latino	1 (5.3%)	1 (10.0%)	0 (0%)	0.99
	Not Hispanic or Latino	18 (94.7%)	9 (90.0%)	9 (100%)	
Education, n (%)	High school	2 (10.5%)	1 (10.0%)	1 (11.1%)	0.17
	Trade school	1 (5.3%)	0 (0%)	1 (11.1%)	
	Associate’s degree	5 (26.3%)	1 (10.0%)	4 (44.4%)	
	Bachelor’s degree	8 (42.1%)	5 (50.0%)	3 (33.3%)	
	Graduate school	3 (15.8%)	3 (30.0%)	0 (0.0%)	
Employment, n (%)	Disabled	2 (10.5%)	1 (10.0%)	1 (11.1%)	0.10
	Full-time	12 (63.2%)	5 (50.0%)	7 (77.8%)	
	Not working	4 (21.1%)	4 (40.0%)	0 (0.0%)	
	Retired	1 (5.3%)	0 (0%)	1 (11.1%)	
MS type, n (%)	PPMS	1 (5.3%)	0 (0%)	1 (11.1%)	0.09
	RRMS	16 (84.2%)	10 (100%)	6 (66.7%)	
	SPMS	2 (10.5%)	0 (0%)	2 (22.2%)	
Reason for OCR start, n (%)	Breakthrough disease activity on previous treatment	8 (42.1%)	5 (50.0%)	3 (33.3%)	0.60
	Convenience	2 (10.5%)	0 (0%)	2 (22.2%)	
	First-line therapy	5 (26.3%)	3 (30.0%)	2 (22.2%)	
	Risk reduction from prior treatment	3 (15.8%)	1 (10.0%)	2 (22.2%)	
	Side effects of prior treatment	1 (5.3%)	1 (10.0%)	0 (0.0%)	

Abbreviations: MS, multiple sclerosis; PPMS, primary progressive multiple sclerosis; RRMS, relapsing–remitting multiple sclerosis; and SPMS, secondary progressive multiple sclerosis. ^1^ Median (interquartile range) or frequency (%); ^2^ Wilcoxon rank sum exact test; and Fisher’s exact test.

**Table 2 medicina-60-00659-t002:** VAS-F and SSS scores by visit.

		V1, Screening		*p*-Val ^2^	V2, Baseline, Randomization, OCR1a		*p*-Val ^2^	V3, OCR1b		*p*-Val ^2^	V4, OCR2		*p*-Val ^2^
		Arm			Arm			Arm			Arm		
		Cetirizine (n = 10, 53%) ^1^	Diphenhydramine (n = 9, 47%) ^1^	Differences between Arms	Cetirizine (n = 10, 53%) ^1^	Diphenhydramine (n = 9, 47%) ^1^	Differences between Arms	Cetirizine (n = 10, 53%) ^1^	Diphenhydramine (n = 9, 47%) ^1^	Differences between Arms	Cetirizine (n = 10, 56%) ^1^	Diphenhydramine (n = 8, 44%) ^1^	Differences between Arms
VAS-F	Fatigue	2.36 (2.50)	3.13 (1.65)	0.30	1.92 (2.13)	4.10 (2.19)	0.03	1.94 (1.12)	3.95 (1.94)	0.04	2.12 (2.12)	3.52 (1.08)	0.09
	Energy	6.02 (2.80)	5.36 (2.34)	0.50	6.28 (2.56)	5.20 (1.35)	0.40	6.08 (1.80)	4.63 (1.98)	0.13	5.82 (2.43)	5.33 (2.66)	0.70
SSS		1.90 (1.60)	2.67 (0.71)	0.02	1.90 (0.88)	3.11 (1.27)	0.20	2.00 (0.67)	3.13 (1.13)	0.30	2.33 (1.00)	2.50 (0.84)	0.03

Abbreviations: VAS-F, visual analog scale for fatigue; SSS, Stanford sleepiness scale. ^1^ Mean (SD); ^2^ Wilcoxon rank sum test.

**Table 3 medicina-60-00659-t003:** MFIS and MSIS-29 scores by visit.

		V1, Screening		V4, OCR2		*p*-Value ^2^
		Arm		Arm		
		Cetirizine (n = 10, 53%) ^1^	Diphenhydramine (n = 9, 47%) ^1^	Cetirizine (n = 10, 56%) ^1^	Diphenhydramine (n = 8, 44%) ^1^	Differences between Arm Means
MFIS	Physical	9.20 (12.4)	15.2 (8.98)	8.90 (12.2)	11.6 (9.24)	0.24
	Cognitive	9.10 (10.8)	14.1 (8.21)	7.60 (9.85)	9.63 (9.40)	0.57
	Psychosocial	2.00 (3.23)	3.11 (2.80)	1.80 (3.08)	2.63 (1.85)	0.15
	Total	20.3 (24.7)	32.4 (15.4)	18.3 (22.2)	23.9 (15.9)	0.32
MSIS-29	MSIS29 Physical	13.7 (25.9)	27.0 (19.9)	6.38 (9.31)	19.7 (25.7)	0.13
	MSIS29 Psychological	16.3 (22.6)	28.8 (21.4)	12.3 (14.1)	13.0 (10.2)	0.90

Abbreviations: MFIS, modified fatigue impact scale; MSIS-29, multiple sclerosis impact scale). ^1^ Mean (SD); ^2^ Analysis of covariance, adjusting for the screening score.

**Table 4 medicina-60-00659-t004:** TSQM scores by visit.

		V2, Baseline, Randomization, OCR1a		*p*-Value ^2^	V3, OCR1b		*p*-Value ^2^	V4, OCR2		*p*-Value ^2^
		Arm			Arm			Arm		
		Cetirizine (n = 10, 53%) ^1^	Diphenhydramine (n = 9, 47%) ^1^		Cetirizine (n = 10, 53%) ^1^	Diphenhydramine (n = 9, 47%) ^1^		Cetirizine (n = 10, 56%) ^1^	Diphenhydramine (n = 8, 44%) ^1^	
TSQM	Global Satisfaction	90.8 (9.99)	88.0 (15.7)	0.50	93.3 (11.7)	88.6 (14.7)	0.60	99.1 (2.67)	91.7 (13.9)	0.20
	Effectiveness	86.7 (11.2)	78.7 (18.2)	0.50	91.7 (14.2)	78.1 (14.0)	0.08	93.5 (10.0)	87.5 (15.6)	0.30
	Side Effects	100 (0)	94.4 (11.0)	0.20	100 (0)	92.7 (10.4)	0.07	96.3 (11.0)	95.8 (6.98)	0.40
	Convenience	96.1 (5.89)	87.6 (13.9)	0.50	95.6 (7.33)	90.3 (11.0)	0.07	95.7 (10.9)	94.4 (7.04)	0.40

Abbreviations: TSQM, treatment satisfaction questionnaire for medication. ^1^ Mean (SD); ^2^ Wilcoxon rank sum test.

## Data Availability

Anonymized data not published in this article will be made available to qualifying investigators upon approval by Providence IRB and administration.

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
