# Peer review of "Comparing the Risk of Infusion-Related Reactions and Tolerability in Patients Given Cetirizine or Diphenhydramine Prior to Ocrelizumab Infusion (PRECEPT)"

_medicina, 2024, doi:10.3390/medicina60040659_

Round 1
Reviewer 1 Report
Comments and Suggestions for Authors
The article presented here is interesting. The study design is solid, and the results are not misleading; they are clearly presented.
I only have minor comments.
I would suggest adding the ethical protocol number in the Materials and Methods sections.
My next comments are about the Discussion section.
I would suggest discussing what would constitute a good scale (sensitive enough) or how to modify the MFIS MSIS-29 to increase sensitivity.
What would you do to enhance the study's power to detect significant differences between groups?
I would also discuss why some patients experience adverse effects.
Finally, I would suggest discussing the limitations of this study and proposing directions for future research.
Thank you
Author Response
See attached word document.

Reviewer 2 Report
Comments and Suggestions for Authors
Author Response
See attached word document
